# Diabetic Foot Care: A Screening on Primary Care Providers’ Attitude and Practice in Riyadh, Saudi Arabia

**DOI:** 10.3390/medicina59010064

**Published:** 2022-12-28

**Authors:** Sultan Alsheikh, Hesham AlGhofili, Reema Alageel, Omar Ababtain, Ghadah Alarify, Nasser Alwehaibi, Abdulmajeed Altoijry

**Affiliations:** 1Division of Vascular Surgery, Department of Surgery, College of Medicine, King Saud University, Riyadh 11322, Saudi Arabia; 2Department of Vascular Surgery, King Salman Heart Center, King Fahad Medical City, Riyadh 12231, Saudi Arabia; 3Division of Vascular Surgery, Department of Surgery, University of Toronto, Toronto, ON 11525, Canada; 4Department of Surgery, King Fahad Specialist Hospital, Buraidah 52366, Saudi Arabia

**Keywords:** diabetic foot, diabetic foot disease, diabetic foot ulcer

## Abstract

*Background and Objectives:* Diabetic foot (DF) disease is one of the myriad complications of diabetes. Positive outcomes are expected through a multidisciplinary approach as provided by primary care providers (PCPs). This study aimed to assess the knowledge of DF and attitude of physicians in primary healthcare settings toward DF diagnosis and prevention in Saudi Arabia. *Materials and Methods:* This observational cross-sectional study used a self-administered questionnaire that was completed by family medicine consultants, residents, and general practitioners working in primary care settings in Riyadh. *Results:* Of the 152 physicians who completed the survey, (43.4%) completed more than 10 h of diabetes continuing medical education (CME) over the past three years. Most (96.1%) PCPs educate patients about foot self-inspection, and only (64.5%) perform foot inspection at every visit in high-risk diabetic foot patients. PCP knowledge about diagnosing and managing diabetic foot infection was suboptimal. Only 53.9% of participants reported performing a probe-to-bone in DF patients with open wounds. *Conclusions:* We identified knowledge and action gaps among PCPs. Physicians had acceptable knowledge about preventive measures. However, deficits were found regarding diagnosing and management of DF infections. We recommend addressing these knowledge gaps by incorporating DF lectures and workshops within family medicine conferences and residency programs.

## 1. Introduction

Diabetes mellitus (DM) is a metabolic disorder that is characterized by prolonged hyperglycemia. A study performed in 2019 estimated that DM international prevalence was 9.3% (463 million people) and by 2045 will increase to 10.9% (700 million people) [1]. Uncontrolled DM may cause different complications, such as retinopathy, nephropathy, neuropathy, vascular abnormalities, foot injuries, and ulcerations [2,3]. Many factors produce DM complications. The rise in glycation plays the major role. Advanced glycation end-products are complex compounds resulting from non-enzymatic glycation. Several publications showed the implication of these end-products in the development of diabetes micro- and macrovascular complications [4]. Primary care providers can utilize skin autofluorescence which is one of the novel techniques that can be used to detect the accumulation of glycation end-products, thus, can aid in the detection and follow-up of diabetes micro complications [5,6]. A study performed in Saudi Arabia reported that 3.3% of diabetic patients were diagnosed with diabetic foot (DF) complications [7]. In the United States, diabetic foot disease (DFD) management costs from 9 to 13 billion dollars a year apart from general DM costs [8,9]. DF ulcers heal poorly due to insufficient blood flow and nerve damage to the feet. DF causes almost two-thirds of all non-traumatic amputations; therefore, DF management is fundamental [10]. As part of the healthcare transformation strategy, the Saudi healthcare system is moving toward a focus on the delivery of care through primary care services. In light of the above-mentioned numbers about the DF comorbidities and treatment cost, and despite the central role of physicians practicing in primary care settings in the future of the healthcare system in the Kingdom, to our awareness, the knowledge of and attitude towards DF patients from primary care providers have not been assessed in the region. Riyadh is the capital of Saudi Arabia and has the majority of ministry of health resources. This study is meant to assess physicians’ knowledge of and attitude toward DF diagnosis and prevention in primary healthcare settings in Riyadh, Saudi Arabia. 

## 2. Methods

### 2.1. Study Design, Subject Selection, and Sample Size

This study was conducted with the approval of the Institutional Review Board of King Fahad Medical City (KFMC), Riyadh, Saudi Arabia and in accordance with international ethical standards (project number: 19-034E). Participants gave their informed consent for the anonymous use of their data. This study was a cross-sectional design using self-administered questionnaires that were distributed between November 2021 and March 2022. The target respondents were physicians who work in primary care settings in Riyadh, Saudi Arabia. Therefore, the sample included family medicine consultants (attendings/staff), family medicine residents, and general practitioners (GPs). In Saudi Arabia, GPs are doctors who completed medical school but did not enroll in any residency program or doctors who completed a master’s degree in clinical medicine and started practicing without completing a residency. The sample size was calculated based on previous study findings, which indicated that the overall knowledge score of family medicine physicians about type 2 diabetes management is 66.6% [11]. Using a standard sample size equation that assumed balanced responses between groups with a confidence interval (CI) level of 95% and a margin of error of 5%, the minimum required sample size was estimated at 344; thus, we invited this number to participate in the study. Four-hundred fifteen primary care centers are located in Riyadh. We used conventional sampling technique in selecting primary care centers. We assigned a number to every primary care center from one to four-hundred fifteen. From our observation, each primary care center has between six to eight physicians. Therefore, we used a random number generator to select fifty numbers to be able to reach our target sample size. The data collectors provided the questionnaire to the participants and then collected them immediately after completion to avoid the possibility of participants looking up topics and to assure the credibility of participants’ replies.

### 2.2. Data Collection Tool

Considering the available guidelines, a self-report questionnaire was constructed with assistance from field experts [12,13]. The questionnaire was tested on a pilot sample of three physicians (n = 3) who were not included in the study, particularly regarding language and questions receptivity. There were no changes required to the adjusted questionnaire. The questionnaire included 24 questions. The questionnaire consisted of multiple-choice questions distributed in three parts. The first part included demographic data such as age, sex, primary care setting type (academic/community), number of continuing medical education (CME) hours spent on diabetes education, and years of experience. The second part included items that assessed knowledge about preventive measures against DF development, such as the frequency of foot inspections and the recommended glycemic control in diabetics. Furthermore, we asked how often they encourage patients to wear specialized therapeutic footwear and whether they provide preventive foot care instructions to them and their families. The final part assessed their understanding and attitudes towards diagnosis and management strategies for patients with diabetic foot infections (DFI). At the end of the survey, we asked them to identify the obstacles to delivering optimal care to DF patients (multiple answers allowed).

### 2.3. Statistical Analysis

The data were analyzed using IBM SPSS 21.0 version statistical software. For nominal variables, data were depicted as frequency and percentage, and for numerical variables, mean ± standard deviation (SD). We used the chi-squared test to compare proportions between groups. Statistical significance was determined at a *p*-value of <0.05 to report the precision of the results. 

## 3. Results

The target number of participants for the study was 344; 152 physicians agreed to fill the questionnaire with a response rate of 152/344 (44.2%). Participants were segregated into three groups: family medicine consultants (31.7%), family medicine residents (44.7%), and GPs (23.7%). Male to female ratio is 0.9:1. More than half of physicians work in community settings (64.5%). The number of practice years differed based on their training level (*p* < 0.001). Less than 50% (43.4%) of the participants spent more than 10 CME hours on diabetes education over the last three years (*p* < 0.001). For further details, please see Table 1.

Respondents were deemed informed about the targets if they chose the responses based on the most recent guidelines [12,13]. A summary of recommendations is presented in Table 2. 

When the participants were asked to self-evaluate their knowledge of therapies and prevention of DF, 71.1% of participants rated themselves as average. Only 19.7% of the participants evaluated themselves as above average. When the participants were asked about interval foot inspections for high-risk diabetic patients, only 64.5% (*p* = 0.04) of them answered “Every visit”. A total of 41.7% of consultants think feet should be inspected less frequently. Most participants (96.1%) educate their patients and their families about preventive foot care. Most family medicine consultants (70.8%) routinely advise their diabetic patients to use specialized therapeutic footwear. Of all participants, only 26.3% of participants picked the correct answer “footwear to high-risk patients”. Most (90.1%) physicians recommend wearing specific therapeutic footwear to aid in the prevention of new or recurrent foot ulcers in high-risk patients with healed DFUs. Most (95.8%) of these physicians were family medicine consultants. Many (88.8%( participants correctly identified the appropriate HbA1c level for those patients as “HbA1c less than 7%”; the majority of correct answers were from family medicine consultants (93.8%). Finally, regarding the ankle-brachial index/toe-brachial index (ABI/TBI) measurement for diabetic patients, only 28.3% chose “when patients reach 50 years of age”, and most of those participants were consultants. A small number of participants (15.1%) never order it. Further details are shown in Table 3.

Almost three-quarters (67.8%) of the physicians rated the percentage of their diabetic patients whom they have evaluated systematically to be at risk for DF as “less than 50%”; most of these were residents (50%; *p* < 0.001). Only 53.9% of participants probe to check for bone exposure in patients with DF infections. Less than three-quarters (63.8%, mostly consultants; *p* < 0.001) request serial plain radiographs of the affected foot to identify any bone abnormalities. More than half (61.1%) of the GPs do not request a serial plain radiograph, and 68.4% choose MRI as their imaging choice when soft tissue abscess is suspected, or a diagnosis of osteomyelitis remains uncertain. Less than 25% (22.4%) of incorrect answers were attributed to computed tomography (CT) scans. Almost 70% (67.8%) of the participants obtain measurements of DF wounds, with family medicine residents being the vast majority (46.6%). Around one-third of the consultants and GPs do not measure wound size. Furthermore, 54.2% of the family medicine consultants follow up with their patients with DF wounds for 2 to 3 weeks. A total of 66.7% of GPs follow them up as needed. Only 45.4% of the participants order daily dressings for DFD patients. Almost 25% (23%) do not manage diabetic foot wounds. In addition, 49.3% of the physicians, primarily consultants (44%), refer patients with diabetic foot wounds to a vascular surgeon. Further details are shown in Table 4.

When we asked the participants about the most important barriers to optimal DF care, in descending order, lack of DF management guidelines was the most agreed upon factor (57.9%) followed by the lack of continued education about the importance of DFD (55.2%), and lack of knowledge of treating physicians about DFD (51.3%). The absence of vascular medicine specialty in Saudi Arabia (5.3%) followed those choices, and finally, lack of specialized DF clinics in primary care settings, and lack of access to services (0.7%, each) were also mentioned (Figure 1). Last, we asked the participants if they could suggest any other barriers, and “no specialties to refer to in some facilities” (0.7%; n = 1), and “poor communication with the other specialties” (0.7%; n = 1) were the two qualitatively suggested answers causing possible obstacles.

## 4. Discussion

DFU is one of the myriad complications of diabetes. Among diabetic patients, the lifetime risk of DFU is 25% [14]. Around 60% of diabetic wounds are infected at presentation [15]. The lower extremity amputation rates in infected ulcers are as high as 28% [16]. Based on a large study performed involving Saudi diabetic patients, the risk of developing DFUs is 2%, and the amputation rate is 1% [7]. Seventeen percent of DFU patients who were presented to a tertiary hospital in Saudi underwent amputation, and this rate has been consistent over the past years [17,18]. Various resources to help manage this condition are available, but it is unknown whether PCPs and patients are aware of or have trouble accessing them. The way toward optimal DF care is a multidisciplinary approach. No clear coordinated multidisciplinary care is present in the region. The way to enact this care is most commonly through PCPs who form the first line of care in this model. Thus, we aimed to identify attitudes and obstacles facing PCPs when providing optimal care for DFU patients. 

In our study, around one-third of the participants (35.5%) reported that they do not perform foot inspections and examinations at every visit for high-risk patients, which introduces a significant chance of missing early DFUs. In a study performed in four European countries, DFU diagnosis was incidentally found during routine examination in 20% of patients [19]. Most participants (96.1%) educate their patients about self-inspection. On the other hand, a study performed involving patients in Riyadh showed that only 33.3% of patients received education about foot care from their PCPs [19]. This difference most likely occurred because PCPs do not usually provide the education themselves but use a diabetes education specialist who is available in most of the primary care centers in Saudi; in the same study, the level of patient knowledge is reported to be good (76.6%). Even with proper education, some patients still do not perform a self-inspection. Even with adequate knowledge, only 28–47% of patients perform foot self-inspection [20]. Overall, studies assessing the benefits of educational intervention at the patient level are few and are considered low-level, showing modest improvement in outcome [21,22]. This finding sheds light on the importance of “every visit” inspection by PCPs. A study performed in Saudi showed an 8% (*p* = 0.3) reduction in amputation rates after implementing patient education programs. Although the small sample size may affect the statistical significance of their study, the clinical outcome is significant. Educating patients and their families about foot care makes empirical sense and is cost-effective. This education should be provided on an annual basis. Other forms of education can be online. This form of education can be accessed anytime and by any family member, which provides consistency and spread of knowledge about foot care [23].

Around 73% of the participants prescribe therapeutic footwear regardless of DFU risk. Well-fitted shoes can decrease callus development, and toe deformity, thus, decreases the risk of DFU [24]. However, therapeutic custom diabetic footwear cannot be recommended over a preventive footcare program in low-risk patients [12]. No difference was reported in a trial evaluating re-ulceration among patients with therapeutic footwear vs. control group [25]. On the other hand, the high prescription rate may cause a false patient perception that this kind of shoe can prevent ulcers, and therefore, the patient becomes less focused on other foot care measures. 

Only 28.3% of PCPs correctly chose referring patients for ABI/TBI testing when the patients reach 50 years of age. Evidence suggests that TBI is useful in predicting not only wound healing but the potential of ulceration. The relation between DFU and peripheral artery disease (PAD) is complex. Mortality in patients with PAD and DFU who undergo amputation is 50% in two years [12]. Poor limb perfusion can result in ulcers, poor healing, and ultimately, amputation. Thus, early identification of PAD should be attempted, and if it contributes to delayed healing or non-healing, it should be treated [26]. Timing is essential for preventing amputation in addition to early referrals, imaging, and regular vascular tests [27]. Furthermore, early identification helps early establishment of the multidisciplinary circle. 

Approximately 50% of the PCPs in our study probe to check for bone exposure. Furthermore, 68.4% order MRIs to rule out osteomyelitis. These percentages reflect concerning practice toward ruling out infections. Probe-to-bone can accurately diagnose DFU osteomyelitis in high-risk patients [28]. MRI is generally considered the best available imaging option for diagnosing osteomyelitis [29]. A study assessed knowledge of medical students concerning foot examination in DF patients and reported good overall knowledge [30]. A smaller number of students were assessed for foot edema and shoe suitability. The study did not ask questions related to ulcer examination or investigating infections. Therefore, it is unclear if knowledge gaps started in medical school. 

Our study had a lower-than-expected response rate (44.2%), which may have affected the strength of its outcomes, and our results were disappointing. Unfortunately, our study results are in-line with previous global publications about diabetes management [31,32,33]. Participants were aware of maintaining adequate blood glucose levels. However, less awareness was observed regarding other aspects of care to improve DFU outcomes, such as adequate wound care and management, early identification of infection, and early restoration of blood flow [13]. Besides the low-quality of life, the cost of DFU care in one hospital in one year in Saudi is estimated to be 661,804.3 SAR (176,481.2 USD) [34,35]. Improving DFU outcomes should be approached using a multilevel approach. Participants in our study lean toward knowledge aspects as barriers rather than availability or accessibility of resources. To our knowledge, no standardized formal course is uniformly included in the core curriculum at all training institutions. Furthermore, conferences concerning wound care are less likely to be attended by PCPs. As seen in our study, more than half of participants have less than 10 CME hours. A myriad of reasons for this finding can be suggested. PCPs may be too busy. Their scope of practice may be broad and as such, the conferences and CME events they choose to attend are related to other disease entities. Interest in dealing with DFUs may also be low. A suggested action is to include lectures and workshops about DFD within the curriculum of family medicine conferences and residency programs. As per randomised controlled trials, educational interventions have been shown to be effective [31,32,33]. E-learning, when combined with a post session exam, can yield desired outcomes [36]. Although beyond the participants of this study, in more than 20% of DFU cases in a study, the PCP did not make the diagnosis, and the nurses played a significant role in the diagnostic process. Therefore, knowledge and education concerning DFUs should be reinforced to nurses as they are in closer contact to patients in in-hospital settings [37]. This study looked at PCPs practices in randomly selected centers in one city in Saudi Arabia. However, the sample is not representative of all Saudi family medicine physicians and GPs due to interregional differences in medical schools and physician training. This study has some limitations. Our survey is in a multiple-choice answer format; thus, it may limit the physicians’ choice to the most appropriate choice rather than their actual answers. Furthermore, the sampling technique was conventional. Thus, there is a risk of selection bias. However, the sample group is not the general population but rather medically trained and licensed physicians. Therefore, the utilization of this technique should not impose significant outcomes alteration.

## 5. Conclusions

Our results can serve as a screening result of the status of DF care in Riyadh, Saudi Arabia. Knowledge about diagnosis of DFD is adequate. However, diagnosis and management of infections, and requestions of appropriate imaging is suboptimal. The majority of participants think barriers concerning acquisition of knowledge are the most limiting factors toward adequate knowledge. These results have significant implications for family medicine program directors, medical schools, and healthcare stakeholders. Access to knowledge is an easy-to-overcome barrier with adequate spread of education during conferences, and through medical schools, residency programs, and E-learning.

## Figures and Tables

**Figure 1 medicina-59-00064-f001:**
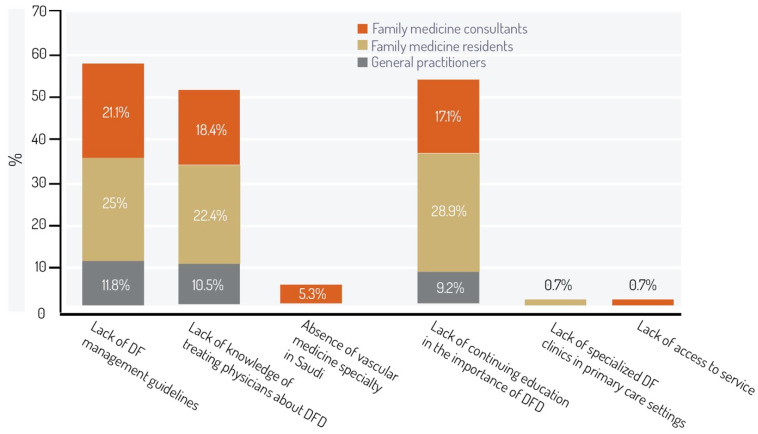
Barriers toward optimal care for diabetic foot patients. DF, diabetic foot; DFD, diabetic foot disease.

**Table 1 medicina-59-00064-t001:** Participants’ demographics and clinical practice characteristics (N = 152).

	Family Medicine Consultants (n = 48)	Family Medicine Residents (n = 68)	General Practitioners (n = 36)	All (n = 152)	*p* Value
Age					
Mean (SD)	43.40 (9.58)	31.54 (7.59)	38.11 (13.09)	36.84 (10.98)	<0.001 *
Sex, n (%)					
Male	29 (60.4)	24 (35.3)	18 (50.0)	71 (46.7)	0.02 *
Female	19 (39.6)	44 (64.7)	18 (50.0)	81 (53.3)	
Setting, n (%)					
Academic center	19 (39.6)	26 (38.2)	9 (25.0)	54 (35.5)	0.32
Community center	29 (60.4)	42 (61.8)	27 (75.0)	98 (64.5)	
Years in practice, n (%)					
<5	3 (6.3)	61 (89.7)	18 (50.0)	70 (46.1)	<0.001 *
5–10	18 (37.5)	7 (10.3)	6 (16.7)	31 (20.4)	
>10	27 (56.3)	0 (0)	12 (33.3)	51 (33.6)	
Number of diabetic Continuing Medical Education (CME) hours completed in past three years *, n (%)					
<5	6 (12.5)	21 (30.9)	14 (38.9)	41 (27.0)	0.07
5–10	7 (14.6)	28 (41.2)	10 (27.8)	45 (29.6)	
>5	35 (72.9)	19 (27.9)	12 (33.3)	66, (43.4)	

CME, continuing medical education; SD, standard deviation; * *p* < 0.05, statistically significant difference.

**Table 2 medicina-59-00064-t002:** Current clinical practice guidelines by the society of vascular surgery in collaboration with the American Podiatric Medical Association and Society for Vascular medicine about the management of diabetic foot (DF).

	Recommendation	Class	Level of Evidence
Prevention of diabetic foot ulceration		1	C
Foot inspection	Annually, and more frequent in high-risk patients		
Foot care patient education	Annually	1	C
Therapeutic footwear	Against (average-risk patients)	2	C
	Recommended (high-risk patients)	1	B
Hb A1c	<7%	2	B
PAD and DFU			
ABI/TBI	50 years of age	2	C
Diagnosis of diabetic foot infection			
DF with open wound	Performing probe to bone	2	C
All patients presenting with new DFI	Serial plain radiographs	2	C
MRI	When soft tissue abscess is suspected, or osteomyelitis diagnosis is uncertain	1	B
Wound care in DFU			
Evaluation interval	1–4-week intervals, with wound measurement	1	C
Interval of dressing change	Daily	1	B

HbA1c, glycated hemoglobin; DFU, diabetic foot ulcer; PAD, peripheral artery disease; ABI, ankle-brachial index; TBI, toe-brachial index; DF, diabetic foot; DFI, diabetic foot infection; MRI, magnetic resonance imaging. Class 1: evidence and/or general agreement that the target or treatment is beneficial. Class 2: conflicting evidence and/or a divergence of opinion, but the weight of evidence favors efficacy. Level B: Randomized or non-randomized clinical trials Level C: Consensus opinion of experts based on clinical experience.

**Table 3 medicina-59-00064-t003:** Participants’ understanding and attitude toward prevention of DF ulceration in diabetic patients.

	Family Medicine Consultants (n = 48)	Family Medicine Residents (n = 68)	General Practitioners (n = 36)	All (n = 152)	*p* Value
Self-assessment knowledge of therapies and prevention of DFI disease, n (%)					
Below average	1 (2.1)	4 (5.9)	9 (25.0)	14 (9.2)	
Average	31 (64.6)	53 (77.9)	24 (66.7)	108 (71.1)	
Above average	16 (33.3)	11 (16.2)	3 (8.3)	30 (19.7)	
How often should high risk diabetic foot patients undergo interval foot inspections? n (%)					
Every visit	28 (58.3)	49 (72.1)	21 (58.3)	98 (64.5)	0.04 *
Every 6 months	7 (14.6)	5 (7.4)	10 (27.8)	22 (14.5)	
Annually	13 (27.1)	14 (20.6)	5 (13.9)	32 (21)	
Do you educate the diabetic patients and their families about preventive foot care? n (%)					
No	0 (0.0)	1 (1.5)	5 (13.9)	6 (3.9)	0.002 *
Yes	48 (100.0)	67 (98.5)	31 (86.1)	146 (96.1)	
In your practice, how often do you advise diabetic patients to use specialized therapeutic footwear? n (%)					
Routinely	34 (70.8)	40 (58.8)	15 (41.7)	89 (58.6)	0.05 *
Rarely	5 (10.4)	8 (11.8)	10 (27.8)	23 (15.1)	
Only high-risk patients	9 (18.8)	20 (29.4)	11 (30.6)	40 (26.3)	
Do you recommend wearing specific therapeutic footwear to aid in the prevention of new or recurrent foot ulcers in high-risk patients with healed DFU?, n (%)					
I do not know	0 (0.0)	3 (4.4)	3 (8.3)	6 (3.9)	0.32
No	2 (4.2)	4 (5.9)	3 (8.3)	9 (5.9)	
Yes	46 (95.8)	61 (89.7)	30 (83.3)	137 (90.1)	
Knowledge about adequate glycemic control in diabetic patients, n (%)					
Fasting blood glucose < 120 mg/dL	1 (2.1)	1 (1.5)	2 (5.6)	4 (2.6)	0.37
HbA1c < 6%	2 (4.2)	6 (8.8)	5 (13.9)	13 (8.6)	
HbA1c < 7%	45 (93.8)	61 (89.7)	29 (80.6)	135 (88.8)	
When do you order ankle-brachial index (ABI) measurement for diabetic patients? n (%)					
Never	6 (12.5)	13 (19.1)	4 (11.1)	23 (15.1)	0.01 *
When they have non-healing wounds	12 (25.0)	15 (22.1)	8 (22.2)	35 (23.0)	
When they have wounds	4 (8.3)	8 (11.8)	5 (13.9)	17 (11.2)	
When they reach 50 years of age	16 (33.3)	19 (27.9)	8 (22.2)	43 (28.3)	
You do it to all diabetic patients.	9 (18.8)	12 (17.6)	3 (8.3)	24 (15.8)	
I do not know	1 (2.1)	1 (1.5)	8 (22.2)	10 (6.6)	

DFI, diabetic foot infection; mg/dL: milligrams per deciliter; HbA1c: glycated hemoglobin; CT: computed tomography; MRI: magnetic resonance imaging. ABI: ankle-brachial index; TBI, toe-brachial index; * *p* < 0.05, statistically significant difference.

**Table 4 medicina-59-00064-t004:** Participants’ collective understanding and attitudes toward diagnosis and management of diabetic foot infections in patients with diabetic foot.

	Family Medicine Consultants (n = 48)	Family Medicine Residents (n = 68)	General Practitioners (n = 36)	All (n = 152)	*p* Value
Among what proportion of your diabetic patients have you evaluated systematically has a risk for the diabetic foot? n (%)					
<50%	25 (52.1)	52 (76.5)	26 (72.2)	103 (67.8)	<0.001 *
>50%	20 (41.7)	13 (19.1)	2 (5.6)	35 (23.0)	
None	3 (6.3)	3 (4.4)	8 (22.2)	14 (9.2)	
In patients with DFI with an open wound, do you do probing to check for bone exposure? n (%)					
I do not know	2 (4.2)	5 (7.4)	1 (2.8)	8 (5.3)	0.36
No	16 (33.3)	27 (39.7)	19 (52.8)	62 (40.8)	
Yes	30 (62.5)	36 (52.9)	16 (44.4)	82 (53.9)	
In patients with a DFI wound, do you request a serial plain radiograph of the affected foot to identify any bone abnormalities? n (%)					
I do not know	0 (0.0)	2 (2.9)	2 (5.6)	4 (2.6)	<0.001 *
No	11 (22.9)	18 (26.5)	22 (61.1)	51 (33.6)	
Yes	37 (77.1)	48 (70.6)	12 (33.3)	97 (63.8)	
For those patients who require additional imaging—particularly when soft tissue abscess is suspected or the diagnosis of osteomyelitis remains uncertain—what type of imaging will you do?, n (%)					
CT scan	10 (20.8)	16 (23.5)	8 (22.2)	34 (22.4)	0.62
Leukocyte or Anti-granulocyte scan	1 (2.1)	2 (2.9)	1 (2.8)	4 (2.6)	
MRI	35 (72.9)	47 (69.1)	22 (61.1)	104 (68.4)	
I do not know	2 (4.2)	3 (4.4)	5 (13.9)	10 (6.6)	
Do you take wound size measurements of DFI wounds, n (%)					
No	14 (29.2)	17 (25.0)	11 (30.6)	42 (27.6)	0.96
Yes	32 (66.7)	48 (70.6)	23 (63.9)	103 (67.8)	
I do not know	2 (4.2)	3 (4.4)	2 (5.6)	7 (4.6)	
How often do you follow up with patients with DFI wounds?, n (%)					
2–3 weeks	26 (54.2)	32 (47.1)	5 (13.9)	63 (41.4)	0.006 *
6–8 weeks	2 (4.2)	3 (4.4)	2 (5.6)	7 (4.6)	
As needed	13 (27.1)	29 (42.6)	24 (66.7)	66 (43.4)	
I do not follow	7 (14.6)	3 (4.4)	5 (13.9)	15 (9.9)	
I do not know	0 (0.0)	1 (1.5)	0 (0.0)	1 (0.7)	
How frequent do you order dressing for patients with DFI wounds? n (%)					
Twice a day	13 (27.1)	12 (17.6)	3 (8.3)	28 (18.4)	0.12
Daily	17 (35.4)	36 (52.9)	16 (44.4)	69 (45.4)	
Every 2 days	8 (16.7)	3 (4.4)	6 (16.7)	17 (11.2)	
I do not manage wounds	10 (20.8)	15 (22.1)	10 (27.8)	35 (23.0)	
I do not know	0 (0.0)	2 (2.9)	1 (2.8)	3 (2.0)	
Where do you order the dressing changes to be done for patients with DFI wounds? n (%)					
At home	9 (18.8)	6 (8.8)	6 (16.7)	21 (13.8)	0.32
In a hospital	11 (22.9)	10 (14.7)	9 (25.0)	30 (19.7)	
In the primary care center	28 (58.3)	51 (75.0)	20 (55.6)	99 (65.1)	
I do not know	0 (0.0)	1 (1.5)	1 (2.8)	2 (1.3)	
To whom may you refer your patients with DFI wounds? n (%)					
General Surgeon	10 (20.8)	24 (35.3)	11 (30.6)	45 (29.6)	0.03 *
Emergency Department	3 (6.3)	13 (19.1)	8 (22.2)	24 (15.8)	
Orthopedic Surgeon	2 (4.2)	1 (1.5)	3 (8.3)	6 (3.9)	
Vascular Surgeon	33 (68.8)	28 (41.2)	14 (38.9)	75 (49.3)	
I do not know	0 (0.0)	2 (2.9)	0 (0.0)	2 (1.3)	

DFU, diabetic foot ulcer; DFI, diabetic foot infection; * *p* < 0.05: statistically significant difference.

## Data Availability

Data available upon request.

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
