# Peer review of "Diabetic Foot Care: A Screening on Primary Care Providers’ Attitude and Practice in Riyadh, Saudi Arabia"

_medicina, 2022, doi:10.3390/medicina59010064_

Round 1

Reviewer 1 Report

This original work shows the role of screening DF diagnosis and prevention. So the paper needs some sentences to add in the introduction:

1- Many factors produce Diabetes Mellitus complications and the increased glycation process relates to the major cause. Advanced glycation end products are complex compounds resulting from non-enzymatique glycation and several publications showed the implication of these products to develop diabetes micro- and macrovascular complications. Authors can add this sentence in the introduction with this reference:

a- Lee J, Yun JS, Ko SH. Advanced glycation end products and their effect on vascular complications in type 2 diabetes complications. Nutrients 2022

2- DM and DF diagnosis and prevention in primary healthcare can use also new non-invasive technologies that can help to follow and to detect early diabetes complications using skin autofluorescence. Authors can add in the introduction this sentence with these references:

a-Boersma HE, van Waateringe RP et al. Skin autofluorescence predicts new cardiovascular disease and mortality in people with type 2 diabetes. BMC Endocr Disord 2021.

b- Varikasuvu SR, Varshney S, Suleker H. Skin autofluorescence as a novel and noninvasive technology for advanced glycation end products in diabetic foot ulcers: a systematic review and meta-analysis. Adv Skin Wound Care 2021.

c- Saky R, Wolffenbuttel BHR et al. Increased skin autofluorescence of advanced glycation end products (AGEs) in subjects with cardiovascular risk factors. International Journal of Diabetes in Developing Countries 2022.

Author Response

Thank you for your input and contribution to our manuscript. The introduction section was commented on by most of the reviewers. We have added those 2 paragraphs to the introduction. Thank you again.

Reviewer 2 Report

Reviewer's comments on the manuscript entitled “Diabetic Foot Care: A Screening on Primary Care Providers Attitude and Practice in Riyadh, Saudi Arabia” (manuscript ID: medicina-2104358).

1. The introduction is too short and therefore, the authors should make it longer. Please include in the introduction information about why your research project is so important based on professional literature and individual factors for the study area or population. What knowledge gaps are filled in? Why is the population studied so important, despite the fact that it only includes doctors working in Riyadh, Saudi Arabia? Why is it worth publishing the findings of a study conducted among these doctors in such a city? The authors are asked to provide reliable justification, supported by professional literature and accurate data. If the authors do not demonstrate that their study and the study group selected are exceptional, then publishing a study conducted in such a heterogeneous population makes no sense and adds little to the state of knowledge, and the study results become only locally-based.

2. The authors should thoroughly describe the method of selection of medical facilities from which the doctors participated in the study were selected. It is recommended that the method of selection be depicted in the figure and described in the text. Were the doctors participated in the study randomly selected? Where the specified criteria applied? The criterion in the selection of the medical facility is not sufficient.

3. Line 62: “Considering the available literature…” What literature? The authors do not refer to any literature references. I would like to read them.

4. The authors’ own questionnaire is used in the study. It is important to thoroughly describe the questionnaire used in the paper. How many questions does it contain? What were the questions about in general? The pilot study of the questionnaire was carried out on a group of 3 doctors but the authors do not indicate any test results. Were any questions changed? Was the questionnaire improved after this pilot study?

5. The sample size amounted to 344 doctors, but only 152 were included in the study. Why? It would be more valuable to describe how the final number of doctors was obtained and to consider the representativeness of the study sample.

6. Why did the authors include Table No 1 in subsection 2.4? What value does it add to the manuscript, given that these are rewritten Scientific Society's guidelines? It is allowed to indicate that the survey questionnaire was constructed on the basis of the guidelines but certainly not in this section. Therefore, such information should not be included in Table No 1, not to mention in the Statistical analysis. On the other hand, subsection 2.4 contains no information on whether the data is normally distributed. The tables include statistical analyses, so which tests were used and why? What statistical significance was adopted? Subsection 2.4. should contain standard information that is characteristic for scientific papers. In this manuscript in section 2.4. these criteria are met only by the first sentence (lines: 85-87).

7. Line 107: “Male to female ratio is: 0.9:1”. What is this sentence for?

8. The results contained in lines 113-130 are unclear. I had to read this part of the manuscript a few times to understand what the authors meant. The authors should take a closer look at this section.

9. The analysis of the study results was carried out using simple statistical tests. The authors should consider conducting the analyses using more advanced tools. This is a suggestion as I am trying to find some strengths in your work.

10. The discussion contains no major errors and therefore, it may remain unchanged. However, no strengths and weaknesses of the study were stressed. Therefore, the authors are asked to indicate strengths and weaknesses of the study.

In summary, I need to conclude that both the article and the study are of low scientific merit. As I do not reject papers without giving the authors the opportunity to provide their comment and make corrections, it is expected that the corrections indicated herein will be implemented. The authors should be aware that if an article with a score close to 3 IF is submitted to a journal, they should carefully read the articles already published in that journal, so as to avoid making such gross errors when preparing their own manuscript.

Author Response

We are grateful for the useful feedback by the reviewers, which has helped us improve the manuscript's quality. We have tried to carefully respond to all points raised by the reviewers and have accordingly modified the manuscript. You will find that we have revised the manuscript with changes that have been highlighted with yellow color indicate where we have made changes based on the reviewers' comments. We have also revised and edited the whole article by professional editing company to make the language clearer and we believe that the paper is much improved.

Point 1: The introduction is too short and therefore, the authors should make it longer. Please include in the introduction information about why your research project is so important based on professional literature and individual factors for the study area or population. What knowledge gaps are filled in? Why is the population studied so important, despite the fact that it only includes doctors working in Riyadh, Saudi Arabia? Why is it worth publishing the findings of a study conducted among these doctors in such a city? The authors are asked to provide reliable justification, supported by professional literature and accurate data. If the authors do not demonstrate that their study and the study group selected are exceptional, then publishing a study conducted in such a heterogeneous population makes no sense and adds little to the state of knowledge, and the study results become only locally-based.

Response 1: Introduction has been improved accordingly. Lines 34-40, and 49-55. Thank you for your input.

Point 2: The authors should thoroughly describe the method of selection of medical facilities from which the doctors participated in the study were selected. It is recommended that the method of selection be depicted in the figure and described in the text. Were the doctors participated in the study randomly selected? Where the specified criteria applied? The criterion in the selection of the medical facility is not sufficient.

Response 2: Lines 77-80: We used a random sampling technique to reach to our targeted sample size. The total number of primary care centers according to the ministry of health in Riyadh is 415. We assigned a number to every primary care center from 1 to 415. From our observation, each primary care center has between 6 to 8 physicians. Therefore, we used a random number generator to select 50 numbers to be able to reach our target sample size. We then reach out to the doctors working in these selected centers and ask them to participate in the study. 

Point 3: Line 62: “Considering the available literature…” What literature? The authors do not refer to any literature references. I would like to read them.

Response 3: Line 85: We changed the wording to “guidelines”. Thank you.  

Point 4: The authors’ own questionnaire is used in the study. It is important to thoroughly describe the questionnaire used in the paper. How many questions does it contain? What were the questions about in general? The pilot study of the questionnaire was carried out on a group of 3 doctors but the authors do not indicate any test results. Were any questions changed? Was the questionnaire improved after this pilot study?

Response 4: Lines 85-100: Data collection tool section: we made changes to the section to meet your constructive recommendations. The questionnaire consisted of 24 questions and was divided into parts. We described those parts briefly. We did not need to make changes to the questionnaire after the pilot study. We clarified that under the same section. We are happy to provide the questionnaire if that is within the journal’s interest.

Point 5: The sample size amounted to 344 doctors, but only 152 were included in the study. Why? It would be more valuable to describe how the final number of doctors was obtained and to consider the representativeness of the study sample.

Response 5: Thank you for your comment. Line 119: “The target number of participants for the study was 344; 152 physicians agreed to fill the questionnaire”. Rest of the physicians whom we reached out to either refused due to time constraints, or because of other unspecified reasons.

Point 6: Why did the authors include Table No 1 in subsection 2.4? What value does it add to the manuscript, given that these are rewritten Scientific Society's guidelines? It is allowed to indicate that the survey questionnaire was constructed on the basis of the guidelines but certainly not in this section. Therefore, such information should not be included in Table No 1, not to mention in the Statistical analysis. On the other hand, subsection 2.4 contains no information on whether the data is normally distributed. The tables include statistical analyses, so which tests were used and why? What statistical significance was adopted? Subsection 2.4. should contain standard information that is characteristic for scientific papers. In this manuscript in section 2.4. these criteria are met only by the first sentence (lines: 85-87).

Response 6: We used the chi-squared test to compare proportions between groups. Statistical significance was determined at a p-value of < 0.05 to report the precision of the results. Based on your input, and reviewer 4’s. We moved table 1 to the results section.

Point 7: Line 107: “Male to female ratio is: 0.9:1”. What is this sentence for?

Response 7: We wanted to clarify that there was no bias toward one sex over the other and that our sample is almost equally distributed in terms of sex.

Point 8: The results contained in lines 113-130 are unclear. I had to read this part of the manuscript a few times to understand what the authors meant. The authors should take a closer look at this section.

Response 8: Thank you for the constructive feedback. We tried to make that part clearer by changing the wording. We hope it is now easier to understand.

Point 9: The analysis of the study results was carried out using simple statistical tests. The authors should consider conducting the analyses using more advanced tools. This is a suggestion as I am trying to find some strengths in your work.

Response 9: Thank you for your constructive feedback. As we try to focus on the main objective of the study, we tried not to complicate the statistical tests used. Further projects will of course include more advanced sampling techniques and advanced tools to evaluate the problem.

Point 10: The discussion contains no major errors and therefore, it may remain unchanged. However, no strengths and weaknesses of the study were stressed. Therefore, the authors are asked to indicate strengths and weaknesses of the study.

Response 10: We appreciate your time in reviewing our manuscript. Strengths and weaknesses are mentioned in lines 377-385. Thank you again for your constructive feedback.

We hope the revised version is now suitable for publication and look forward to hearing from you in due course. 

Sincerely, 

Reviewer 3 Report

This cross-sectional observational study aimed to evaluate the knowledge of the diabetic foot and the attitudes of physicians in primary health care settings towards the diagnosis and prevention of this complication of diabetes in Saudi Arabia. To achieve this goal the authors used a self-administered questionnaire that was completed by family medicine consultants, residents, and general practitioners working in primary care settings in Riyadh. Data showed that Physicians had acceptable knowledge about preventive measures but deficits were found regarding diagnosing and management of Diabetic Foot infections.

The topic is interesting and the manuscript is clear and well-structured.

Title and abstract represent an accurate description of the study.

The introduction needs to be improved.

Lines 87-89 this paragraph and table 1 should be moved into the results section

Line 98 there is no level A in the table

Line 107 please remove the parenthesis at the beginning of the sentence

Table 2, table 3 and table 4 please report the statistically significant differences in the tables’ legend

Line 126 please add “ after 7%

Line 163 please replace Figure I with Figure 1

Please review the punctuation throughout the text.

Line 262 please replace “selectin” with “selection”.

Author Response

The topic is interesting and the manuscript is clear and well-structured.

Title and abstract represent an accurate description of the study.

Thank you for taking the time to review our manuscript, we will provide point-by-point responses to your kind comments.

The introduction needs to be improved.

We tried to improve the introduction by adding sentences about the pathophysiology and the importance of our research. We hope it is now satisfactory.

Lines 87-89 this paragraph and table 1 should be moved into the results section

Line 98 there is no level A in the table

Line 107 please remove the parenthesis at the beginning of the sentence

Table 2, table 3 and table 4 please report the statistically significant differences in the tables’ legend

Line 126 please add “ after 7%

Line 163 please replace Figure I with Figure 1

Please review the punctuation throughout the text.

Line 262 please replace “selectin” with “selection”.

We have made changes based on the comments. We have also revised and edited the whole article by a professional editing company to make the language clearer and we believe that the paper is much improved.